# Sodium-Glucose Transporter-2 Inhibitors (SGLT2i) and Myocardial Ischemia: Another Compelling Reason to Consider These Agents Regardless of Diabetes

**DOI:** 10.3390/ijms26052103

**Published:** 2025-02-27

**Authors:** Francesco Piccirillo, Matteo Lanciotti, Annunziata Nusca, Lorenzo Frau, Agostino Spanò, Paola Liporace, Gian Paolo Ussia, Francesco Grigioni

**Affiliations:** 1Fondazione Policlinico Universitario Campus Bio-Medico, Via Alvaro del Portillo, 200, 00128 Roma, Italy; f.piccirillo@unicampus.it (F.P.); matteo.lanciotti@unicampus.it (M.L.); lorenzo.frau@unicampus.it (L.F.); agostino.spano@unicampus.it (A.S.); g.ussia@policlinicocampus.it (G.P.U.); f.grigioni@policlinicocampus.it (F.G.); 2Research Unit of Cardiovascular Sciences, Department of Medicine and Surgery, Università Campus Bio-Medico di Roma, Via Alvaro del Portillo, 21, 00128 Roma, Italy

**Keywords:** cardiovascular disease, sodium-glucose cotransporter-2 inhibitor (SGLT2i), myocardial ischemia, atherosclerosis

## Abstract

In recent years, the introduction of sodium-glucose transporter-2 inhibitors (SGLT2is) marked a significant advancement in the treatment of cardiovascular disease (CVD). Beyond their known effects on glycemic control and lipid profile, SGLT2is demonstrate notable benefits for cardiovascular morbidity and mortality, regardless of diabetic status. These agents are currently recommended as first-line therapies in patients with heart failure, both with reduced and preserved ejection fraction, as they improve symptoms and reduce the risk of hospitalization. While several studies have demonstrated that SGLT2is can reduce the incidence of major adverse cardiovascular events (MACEs), the true impact of these agents on atherosclerosis progression and myocardial ischemia remains to be fully understood. A global beneficial effect related to improved glycemic and lipid control could be hypothesized, even though substantial evidence shows a direct impact on molecular pathways that enhance endothelial function, exhibit anti-inflammatory properties, and provide myocardial protection. In this context, this narrative review summarizes the current knowledge regarding these novel anti-diabetic drugs in preventing and treating myocardial ischemia, aiming to define an additional area of application beyond glycemic control and heart failure.

## 1. Introduction

Cardiovascular disease (CVD), especially myocardial infarction (MI), represents the most common cause of mortality and morbidity worldwide [1]. Significant efforts have been made in primary prevention to reduce cardiovascular risk factors by promoting a healthy lifestyle and the use of drugs that help to prevent plaque formation and progression [2]. Concordantly, in patients with known coronary artery disease (CAD), the use of specific medications such as lipid-lowering agents, antithrombotic drugs, anti-inflammatory therapy, and neurohormonal agents has been associated with a reduction in recurrent major adverse cardiovascular events (MACEs) at long-term follow-up [3].

The introduction of sodium-glucose cotransporter 2-inhibitors (SGLT2is) marked a significant advancement in the management of CVD. Several studies proved the beneficial effects on cardiovascular (CV) morbidity and mortality, regardless of the presence of diabetes [4]. Specifically, several randomized clinical trials highlighted the efficacy of SGLT2is in reducing the risk of CV death or hospitalization due to heart failure [5,6]. As result, current European Society of Cardiology (ESC) guidelines recommend empagliflozin and dapagliflozin as first-line therapies in patients with heart failure [7]; however, their role in CAD and myocardial ischemia remains undefined. As result, SGLT2is are not yet standard in ischemia or atherosclerosis management, where statins, beta-blockers, and antiplatelets remain first-line therapies.

In recent years, the widespread use of SGLT2is demonstrated a range of beneficial effects, extending beyond their well-established efficacy in diabetes mellitus and heart failure to encompass various other cardiovascular conditions. Notably, clinical evidence showed that SGLT2i therapy is associated with a significant reduction in the incidence of atrial fibrillation and/or atrial flutter when compared to placebo [8,9]. Notably, this protective effect appears to be particularly pronounced in patients with HF with reduced ejection fraction [10]. Similarly, the use of SGLT2is is related to lower risk of sudden cardiac death, albeit only in heart failure patients [10].

Moreover, therapy with SGLT2is is associated with a reduction in MACEs in patients with type 2 diabetes mellitus (T2DM) [5,11,12,13]. Beyond their known action on glycemic control, these beneficial effects could be explained by an ameliorated lipid profile [4], in addition to the antioxidant, anti-inflammatory, and antithrombotic properties of this class of agents [14]. However, there is insufficient data regarding the impact of SGLT2is on myocardial ischemia, as only a limited number of clinical trials have investigated their role in patients with CVD or acute coronary syndrome (ACS). Furthermore, the mechanisms by which these agents influence atherogenesis, plaque progression, and instability remain unclear.

On these bases, this review aims to summarize the current knowledge regarding SGLT2is in preventing and treating coronary atherosclerosis and myocardial ischemia, thus defining an additional possible area of application beyond glycemic control and heart failure. Thus, we explore the benefit of these agents for cardiovascular risk factors and molecular pathways that induce atherosclerosis progression.

Details regarding the research methods are described in Appendix A.

## 2. Atherogenesis and SGLT2 Inhibitors

The term atherogenesis refers to a chronic disease that affects the walls of large and medium-sized arteries, resulting in the formation of atherosclerotic plaques. This condition can lead to serious cardiovascular events, such as myocardial infarction, strokes, and peripheral artery disease. It represents a complex and progressive process characterized by the interaction of several factors, such as lipid deposition, inflammation, and vascular injury, in which both non-modifiable and modifiable risk factors are involved as initiators and promoters [15]. In the following sections, we will examine the role of SGLT2is in relation to various risk factors, as evidenced by multiple clinical trials, with a focus on mitigating their potential pro-atherogenic effects (Figure 1).

### 2.1. Hypertension

Hypertension is closely linked to atherogenesis through several mechanisms. Indeed, high blood pressure contributes to both the initiation and progression of atherosclerosis, inducing endothelial damage and dysfunction, causing cholesterol accumulation in arterial walls, favoring the development of a chronic inflammatory environment, and promoting smooth muscle cell proliferation and vascular remodeling. Notably, hypertension is associated with the development of atherosclerosis on a continuous scale, rather than a binary one of normotension versus hypertension. This means that even modest changes in blood pressure (BP) can be associated with significant reduction in the incidence of CVD [16,17]. In the past years, the widespread use of SGLT2is for treating T2DM and heart failure enabled several retrospective analyses of clinical trials examining their effects on BP. These trials showed that SGLT2is can lower both diastolic and systolic BP, though only modestly [18,19]. Importantly, the observed BP-lowering effects of SGLT2is were not limited to the initial phase of treatment but were sustained throughout the follow-up period, demonstrating a long-term impact on both diastolic and systolic BP [20]. Additionally, it is also important to emphasize that blunted nocturnal BP dipping is related to increased CVD [21], and alterations of the circadian BP rhythm are common among patients with T2DM and hypertension. Some studies demonstrated that the circadian BP rhythm is maintained in hypertensive patients treated with SGLT2is [22,23,24]. The SHIFT-J study specifically investigated the effects of canagliflozin on nocturnal home blood pressure in patients with poorly controlled T2DM and nocturnal blood hypertension [25]. The trial found that the addition of canagliflozin to standard antihyperglycemic therapy resulted in a modest reduction in nocturnal systolic blood pressure, while significantly lowering morning and evening home systolic blood pressure and serum NT-proBNP levels compared to other antihyperglycemic agents. Moreover, the SACRA study [26] investigated changes in blood pressure with the use of empagliflozin in addition to existing antihypertensive therapy and showed a significant reduction of blood pressure in patients with uncontrolled nocturnal hypertension. The observed differences in the reduction of diurnal and nocturnal blood pressure induced by SGLT2is could be explained by circadian variations in sympathetic tone and kidney function, which tend to lower blood pressure during the nighttime [12]. Importantly, within the broader context of myocardial ischemia benefits, it is noteworthy that the blood pressure-lowering effects of SGLT2is do not come with a corresponding increase in heart rate. Therefore, the reduction in blood pressure, without an associated increase in heart rate, points to a possible decrease in sympathetic nervous system (SNS) activity. Increasing research indicates that SGLT2 inhibition may lower sympathetic nerve output, suppress the production of tyrosine hydroxylase, and reduce norepinephrine turnover in brown fat [27,28,29]. Furthermore, it has been suggested that that reduced SNS activity may stem from decreased renal stress, which leads to the suppression of renal afferent sympathetic signals [30].

The mechanisms by which SGLT2is reduce blood pressure extend beyond decreased sympathetic nervous system activity, although not fully understood, are:
-*Osmotic diuresis and natriuresis.* The reduction in blood pressure induced by SGLT2i therapy is mainly related to natriuresis and osmotic diuresis due to both glucosuria and natriuresis, which lead to a subsequent decrease in preload [31,32]. Specifically, osmotic diuresis causes early decreases in blood pressure, whereas natriuresis contributes to more sustained longer-term reductions [33].-*Local inhibition of the renin-angiotensin-aldosterone system*. This mechanism is due to the secondary effect of increasing delivery of sodium to the juxtaglomerular apparatus during SGLT2 inhibition [34].-*Reduction in body weight*. Excess weight is associated with higher mean BP. Previous studies have shown that even moderate weight loss can lead to a decrease in BP for both hypertensive and non-hypertensive patients [35]. SGLT2is have demonstrated a significant effect on body weight reduction [36,37,38]. Specifically, in the initial weeks of treatment, weight loss related to SGLT2is appears to result mainly from plasma volume contraction [39]. However, long-term studies also reported a reduction in fat tissue mass, potentially due to the approximately 200 Kcal/day caloric loss related to the increased urinary glucose excretion [18,40,41].-*Decrease in arterial stiffness and vascular resistance*. This effect is attributed to the ability of SGLT2is to reduce endothelial cell activation and promote direct vasorelaxation [42,43].

### 2.2. Diabetes

A comprehensive analysis of the effects of SGLT2is on diabetes lies beyond the scope of this paper. Diabetes is a well-known cardiovascular risk factor, strongly related to atherosclerosis, due to chronic hyperglycemia, insulin resistance, and metabolic disturbances, such as endothelial dysfunction, inflammation, oxidative stress, and a pro-coagulant state, which promotes the formation of arterial plaques, their progression, and instability [44]. The effective management of blood glucose is critical for reducing the risk of atherosclerosis in individuals with diabetes, as sustained hyperglycemia contributes significantly to the pathophysiological processes underlying the development of atherosclerotic lesions [14]. SGLT2is were created as anti-diabetic drugs based on their mechanism of action characterized by the inhibition of sodium-glucose cotransporter 2 (SGLT2) in the kidney’s proximal tubules; this transporter is responsible for reabsorbing about 90% of glucose in urine back into the bloodstream. By inhibiting this transporter, SGLT2i medications promote glucose excretion in urine (glucosuria), lowering blood glucose levels.

### 2.3. Hyperlipidemia

Hyperlipidemia represents a condition characterized by elevated levels of lipids in the blood, such as cholesterol and triglycerides. In this setting, high levels of low-density lipoprotein (LDL) cholesterol could accumulate in the walls of large and medium-sized arteries, including coronary arteries. This accumulation can lead to oxidative modification and macrophage internalization, resulting in the formation of atherosclerotic plaques. Hence, proper lipid management is essential for mitigating the risk of atherosclerosis and its associated cardiovascular complications.

SGLT2is have been proven to act on lipid metabolism through multiple mechanisms, such as the modulation of lipid uptake, synthesis, and mobilization of these molecules. These agents affect key pathways in the liver and adipose tissues that are involved in fat metabolism, causing changes in fat distribution. Specifically, the glucosuria induced by SGLT2is induces a “fasting state” in which energy production shifts toward lipid utilization via beta-oxidation. Moreover, the glucosuria enhances insulin sensitivity, leading to reduced hepatic synthesis and increased catabolism of triglyceride-rich lipoproteins [4]. Furthermore, the use of agents such as empagliflozin and canagliflozin mitigates hepatic steatosis, attenuates hepatic lipogenesis, and stimulates lipolysis within a diabetic context. Additionally, SGLT2i therapy exerts antioxidative effects that reduce lipid peroxidation [4]. Empagliflozin specifically may downregulate the expression of glucose transporter type 4 (GLUT 4) in abdominal adipose tissue, enhancing lipid mobilization and attenuating lipid storage to reduce glycerol synthesis [45]. Similarly, dapagliflozin has been demonstrated to facilitate free fatty acid mobilization and transport, preventing hepatic lipid accumulation [46]. However, data on the effects of SGLT2is on circulating LDL cholesterol levels are conflicting. Some studies have reported an increase in LDL levels with these agents, which may be attributed to reduced LDL receptor expression on hepatocyte surfaces, resulting in decreased clearance of circulating LDL cholesterol. Conversely, other studies have indicated a potential decrease in LDL cholesterol following empagliflozin therapy, likely due to its inhibitory effects on genes associated with cholesterol uptake and biosynthesis [4,47]. This discrepancy could be related to differences in study populations, dosing regimens, or the specific characteristics of individual SGLT2i agents. Most studies conducted to date have demonstrated that SGLT2i therapy induces a raise in LDL and HDL cholesterol levels (and, consequently, in total cholesterol) and a reduction in triglycerides. The shady effects of SGLT2is on lipid metabolism could explain their relative lower benefits for cardiovascular events compared to other anti-diabetic drugs, such as glucagon-like peptide-1 receptor agonists or dipeptidyl peptidase-4 inhibitors. However, these findings were not statistically significant in any case. Further investigation is necessary to better understand the underlying mechanisms contributing to these divergent effects on LDL cholesterol. These conflicting findings emphasize the complexity of how SGLT2is influence lipid metabolism and highlight the need for more comprehensive studies to clarify their impact on lipid profile and cardiovascular risk.

### 2.4. Hyperuricemia

Hyperuricemia is characterized by elevated levels of serum uric acid, which have increasingly been associated with the pathogenesis of atherosclerosis. The relationship between high levels of uric acid and atherosclerosis involves several biochemical and physiological mechanisms, such as the promotion of oxidative stress and inflammatory responses, the activation of monocytes and macrophages, and the induction of endothelial dysfunction, partly through inhibition of nitric oxide synthase, vascular smooth muscle cell proliferation and migration, and increased vascular stiffness [48,49]. Hyperuricemia also interferes with lipid metabolism, contributes to insulin resistance, and leads to renal dysfunction, all of which further exacerbate the atherosclerotic process [50,51,52]. The association between hyperuricemia and atherosclerosis suggests that lowering uric acid levels may represent a potential therapeutic strategy for lowering cardiovascular risk. SGLT2is, particularly empagliflozin and canagliflozin, have demonstrated a significant ability to lower uric acid levels in different clinical trials, systematic reviews, and meta-analyses [53,54,55]. This effect is likely related to the increased renal glucosuria induced by SGLT2 inhibition. In this regard, these drugs promote glucose excretion through urine, enhancing osmotic diuresis, which consequently leads to the renal excretion of uric acid. Furthermore, this mechanism may involve competitive inhibition at the level of renal urate transporters, promoting urate clearance [56,57].

### 2.5. Hyper-Omocysteinemia

High levels of homocysteine have been strongly related to both the development and progression of atherosclerosis. Several mechanisms mediate this association, including the induction of oxidative stress, endothelial dysfunction, smooth muscle cell proliferation, and chronic inflammation. Moreover, hyper-homocysteinemia contributes to plaque instability and promotes a pro-thrombotic state by enhancing platelet activation and increasing fibrinogen levels, which represent critical factors in clot formation. Consequently, these mechanisms can lead to both the initiation and progression of atherosclerotic plaque formation, promoting plaque instability and raising the risk of cardiovascular events such as myocardial infarction and stroke. Hence, lowering homocysteine levels could provide cardiovascular benefits, especially in patients with hypertension, insulin resistance, and other metabolic disorders [58]. SGLT2is have been shown to decrease homocysteine levels significantly. Even though the precise pathways that relate the use of SGLT2is to the reduction in homocysteine levels remain to be clarified, it is hypothesized that these drugs may act on hyper-homocysteinemia through a combination of mechanisms, including improved insulin sensitivity, decreased systemic inflammation and oxidative stress, and enhanced renal function [59]. In particular, a recent study by Xu et al. [59] demonstrated that dapagliflozin significantly reduced serum homocysteine levels in patients with hypertension and insulin resistance, showing that this drug could be a promising therapeutic option in reducing cardiovascular risk in hypertensive patients with homocysteine levels ranging from 10 to 15 μmol/L.

### 2.6. High Plasminogen Activator Inhibitor-1 (PAI-1)

Plasminogen activator inhibitor-1 (PAI-1) is a key regulator of fibrinolysis; this protein inhibits tissue plasminogen activator (tPA) and urokinase-type plasminogen activator (uPA), which are responsible for the conversion of plasminogen to plasmin. Plasmin is crucial for breaking down fibrin clots. Moreover, elevated PAI-1 levels promote vascular inflammation and endothelial dysfunction, which are considered key processes in the formation and progression of atherosclerotic plaques. Thus, increased PAI-1 concentrations both enhance the atherosclerotic process and impair the fibrinolytic system, reducing the breakdown of fibrin, and thereby promoting thrombus formation and persistence. The combination of impaired fibrinolysis, plaque vulnerability, and increased inflammation elevates the risk of acute thrombotic events, such as myocardial infarction and stroke, particularly in individuals with underlying insulin resistance, metabolic syndrome, or other cardiovascular risk factors [60]. PAI-1 is mostly synthesized in the liver and endothelium but also in adipose tissue; obesity and excess visceral fat are indeed strongly associated with elevated PAI-1 levels due to the increased secretion from adipocytes. Furthermore, elevated insulin levels are known to increase PAI-1 production through the insulin’s stimulatory effects on adipocytes. Empagliflozin and dapagliflozin demonstrate a potential impact on modulating PAI-1 levels through several interconnected mechanisms, improving insulin sensitivity and glycemic control, reducing inflammation and oxidative stress, promoting weight loss and decreasing visceral adiposity, and enhancing renal function [61,62]. Collectively, these effects contribute to the reduction in PAI-1 levels in the bloodstream, leading to a mitigation of cardiovascular risk, especially in patients with T2DM and metabolic syndrome.

Table 1 schematizes the effects of SGLT2is on cardiovascular risk factors associated with atherosclerosis.

## 3. Vascular Function and SGLT2 Inhibitors

### 3.1. Endothelial Dysfunction

The endothelium comprises a monolayer of endothelial cells that forms the inner cellular lining of blood vessels, being a key factor in the homeostasis of vascular health and actively contributing to regulating vasodilation, blood flow, inflammation, and coagulation [63]. Endothelial dysfunction is a pathological condition characterized by the loss of normal endothelial function, resulting in compromised vascular homeostasis, which leads to the pathogenesis of various cardiovascular diseases, including atherosclerosis and hypertension [64,65].

When endothelial dysfunction occurs, different mechanisms are negatively affected:
-*Nitric oxide (NO) production*. In endothelial dysfunction, NO synthesis is significantly reduced, causing impaired vasodilation and vasoconstriction, which elevates blood pressure and decreases blood flow [66].-*Inflammatory regulation*. The production of inflammatory mediators and cell adhesion molecules increases, leading to a pro-inflammatory environment that attracts immune cells, including macrophages, which adhere to vessel walls and initiate the formation of atherosclerotic plaques [67].-*Reactive oxygen species (ROS) production.* In endothelium dysfunction, oxidative stress levels rise, causing a higher production of ROS, which damages endothelial cells and oxidizes LDL cholesterol, a key process in plaque progression [68].-*Barrier function.* Under normal circumstances, the endothelium works as a selective barrier between the bloodstream and the wall of blood vessels; when the endothelium goes through a dysfunctional state, the integrity of this barrier is impaired, allowing lipids and inflammatory cells to infiltrate vessel walls and accelerate atherogenesis [69].-*Antithrombotic effect*. The physiological antithrombotic properties of the endothelium are compromised, leading to a pro-thrombotic state that increases the possibility of thrombotic events such as myocardial infarction and stroke [70].

SGLT2is have demonstrated a potential role in ameliorating endothelial dysfunction through multiple mechanisms. In a study conducted by Gaspari et al. [71], the effects of SGLT2is on human vascular endothelial cells were examined under conditions of tumor necrosis factor-α (TNFα) stimulation or hyperglycemia. The study findings indicated that SGLT2 inhibitors can reduce the expression of key inflammatory and adhesion molecules, including intercellular adhesion molecule-1 (ICAM-1), vascular cell adhesion molecule-1 (VCAM-1), PAI-1, and nuclear factor kappa B (NF-κB). By decreasing the expression of these vascular adhesion molecules, chronic administration of SGLT2is may reduce macrophage infiltration into the vessel walls, thereby diminishing the formation and progression of atherosclerotic plaques. Additionally, in this trial, dapagliflozin exerted vasoprotective effects by promoting endothelium-independent vasorelaxation.

One important pathway is the SIRT1/eNOS signaling pathway, which plays a key role in maintaining endothelial health and overall cardiovascular function. SIRT1, through its deacetylase activity, regulates eNOS, which increases nitric oxide production, promoting vasodilation and protecting against endothelial dysfunction [72,73]. Clinically, activating SIRT1 has been linked to better vascular health in conditions such as hypertension, atherosclerosis, and diabetes [74,75,76]. Previous studies showed that compounds like resveratrol can activate SIRT1 and showed potential benefits in both preclinical and clinical settings [77,78,79]. Another pivotal signaling pathway involved in vascular and metabolic health is the AMPK pathway, which regulates cellular energy homeostasis [80]. AMPK activation has been found to enhance insulin sensitivity and safeguard against endothelial dysfunction, particularly in diabetic patients [81]. Furthermore, a recent study by Uthman et al. [82] demonstrated that empagliflozin and dapagliflozin can reduce endothelial dysfunction and cellular aging by activating the SIRT1/eNOS signaling pathway. This activation resulted in improved NO bioavailability, reduced ROS production, and decreased TNFα-induced endothelial inflammation in vitro. These results suggest that SGLT2is may play a protective role in maintaining endothelial function by modulating the key molecular processes involved in oxidative stress and inflammation. Moreover, empagliflozin showed significant effects on microvascular complications associated with diabetes mellitus by modulating the SIRT1/AMPK pathway within mitochondrial networks [83]. Indeed, through the activation of this pathway, empagliflozin reduces oxidative stress and suppresses inflammatory responses, thereby contributing to a decrease in microvascular damage, minimization of coronary microvascular injury, and attenuation of cardiomyopathy development [83].

Empagliflozin also supports the preservation of microvascular barrier integrity and homeostasis by promoting eNOS phosphorylation and facilitating endothelium-dependent vasorelaxation, which enhances micro-vessel density and tissue perfusion [84,85]. Finally, empagliflozin ameliorates microvascular dysfunction through interactions between cardiac microvascular endothelial cells and cardiomyocytes, further supporting cardiovascular microvascular health [86,87]. These findings suggest that empagliflozin may effectively target pathways that are critical to microvascular protection. Several studies also demonstrated that SGLT2is inhibited the progression of atherosclerotic lesions associated with vascular aging in vitro by suppressing the proliferation and migration of vascular smooth muscle cells (VSMCs). This effect was dose dependent for canagliflozin, but it required substantially higher concentrations for empagliflozin and dapagliflozin to achieve similar outcomes [88,89]. Moreover, empagliflozin was proven to determine a modulation in the vascular response to acetylcholine through its glucose-lowering effects, enhancing endothelial function [90].

Table 2 schematizes the effects of SGLT2 inhibitors on different mechanisms of endothelial dysfunction and vascular health.

In conclusion, SGLT2is could significantly ameliorate endothelial dysfunction through several mechanisms, including lowering inflammatory activation, promoting direct vasorelaxation, modulating the activity of perivascular progenitor cells, and maintaining the integrity of the microvascular barrier (Figure 2). Through these combined effects, SGLT2 inhibitors represent a promising therapeutic approach for addressing the multifaceted aspects of endothelial dysfunction, potentially reducing the risk of cardiovascular complications.

### 3.2. Arterial Stiffness

With advancing age, the content of elastin in the arterial walls gradually decreases, leading to a significant reduction in vascular elasticity, a condition known as arterial stiffening [91,92]. This stiffening arises from the natural degradation of elastin fibers due to prolonged mechanical stress, which induces a chronic inflammatory environment. This inflammatory state promotes vascular calcification processes that further contribute to arterial rigidity [93]. Additionally, medial degeneration within the vascular wall is a key driver of arterial stiffness [94]. Within this context, endothelial dysfunction plays a significant role both as a contributing factor to and as a result of arterial stiffness [95]. Arterial aging and stiffness are recognized as independent risk factors in the pathogenesis of atherosclerosis and, consequently, in the development of various cardiovascular diseases, including coronary artery disease, heart failure, and stroke [96]. Furthermore, some studies indicated that individuals with an estimated vascular age “younger” than their chronological age exhibit a significantly lower risk of cardiovascular events than those whose vascular age aligns with their chronological age, even when both groups exhibit a similar prevalence of cardiovascular risk factors [97]. Arterial stiffness promotes cardiovascular disease primarily by increasing pulse wave velocity, pulse pressure, and mean arterial pressure [98,99]. These parameters have been demonstrated to have both prognostic and therapeutic value in predicting cardiovascular risk [100,101]. The impact of arterial stiffness on cardiovascular health is further influenced by comorbid conditions such as hypertension, diabetes mellitus, dyslipidemia, and obesity, which accelerate the progression of arterial stiffening. This creates a bidirectional relationship, in which arterial stiffening not only results from these cardiovascular risk factors but also actively contributes to the persistence and worsening of hypertension [102]. These interactions underscore the importance of addressing arterial stiffness in the context of comprehensive cardiovascular risk management.

The pleiotropic effects of SGLT2is in blood pressure reduction, weight loss, decreasing hyperuricemia, and ameliorating glycemic control (as discussed in the previous section “Atherogenesis and SGLT2 inhibitors”) collectively contribute to attenuating vascular aging and stiffness, thereby providing cardiovascular protection. As previously discussed, SGLT2is can reduce inflammatory mediators and reactive oxygen species (ROS) production, thereby inhibiting medial degeneration and mitigating vascular calcification and the subsequent development of arterial stiffness [103]. Another important role of SGLT2is in mitigating arterial stiffness is attributed to their effects on endothelial dysfunction, already discussed in the previous paragraph (“Endothelial dysfunction”). Moreover, SGLT2is show beneficial effects in slowing the progression of and improving arterial stiffness by reducing the pulse wave velocity and pulse pressure. Specifically, in their study on thoracic smooth muscle cells from rabbits, Li et al. [104] observed that dapagliflozin promoted vasodilation through the activation of voltage-dependent potassium channels and engagement of the protein kinase G (PKG) signaling pathway. These mechanisms indicate that dapagliflozin exerts its vasodilatory effects by modulating ion channels and intracellular signaling cascades, highlighting its potential role in vascular smooth muscle relaxation. In addition, in their study, Bosch et al. [105] reported significant improvements in various vascular parameters following the administration of empagliflozin in patients with T2DM. Specifically, they observed reductions in central systolic blood pressure, central pulse pressure and wave amplitude. These changes led to the beneficial effect of empagliflozin on arterial stiffness, suggesting that this pharmacological intervention may contribute to enhanced vascular function and overall cardiovascular health. Santiago et al. [106] evaluated the impact of dapagliflozin on arterial stiffness in patients with T2DM by measuring carotid-femoral pulse wave velocity. Their findings demonstrated a significant decrease in pulse wave velocity; however, these changes did not correlate significantly with other clinical parameters, including blood pressure, blood glucose concentrations, and body weight. In another study, Bekki et al. [107] reported that switching from DPP-4 inhibitors to tofogliflozin led to improvements in arterial stiffness in T2DM patients, as measured using the cardio-ankle vascular index. Additionally, Katakami et al. [108] provided further evidence of tofogliflozin’s efficacy, showing a reduction in arterial stiffness reflected by decreased brachial-ankle pulse wave velocity in diabetic individuals without known cardiovascular disease. These studies suggested that tofogliflozin effectively slowed the progression of arterial stiffening when compared to conventional treatment options. Furthermore, Jung et al. [109] investigated the effects of combined therapy with empagliflozin and linagliptin versus the combination of metformin and linagliptin. Their results revealed a significant reduction in pulse wave velocity values in the empagliflozin group, suggesting a significant improvement in vascular function associated with this therapeutic regimen.

In summary, SGLT2is may have beneficial effects on arterial stiffness by addressing multiple cardiovascular risk factors, promoting vasorelaxation and improving endothelial function (Table 3). These agents facilitate a reduction in parameters associated with vascular rigidity, thus contributing to enhanced cardiovascular health [43].

## 4. Plaque Instability and SGLT2is

The concept of plaque instability refers to the condition in which an atherosclerotic plaque is prone to rupture or erosion, with consequent exposure of pro-thrombotic components that can lead to thrombus formation and potential occlusion of the affected artery. The main factors that contribute to plaque instability are the composition of the plaque, chronic inflammation, macrophage activity, oxidative stress, and shear stress [110]. By targeting these mechanisms, SGLT2is demonstrate significant potential in reducing the likelihood of plaque rupture or erosion, making them promising agents not only in preventing the development and progression of atherosclerotic plaques but also in enhancing their stability.

### 4.1. Plaque Composition

The structural and cellular composition of atherosclerotic plaques largely determines their stability. A stable plaque typically features a dense, thick fibrous cap and a smaller, more organized lipid core. By contrast, a vulnerable or unstable plaque is characterized by a thin fibrous cap, a large lipid-rich necrotic core, and increased inflammatory cell content [111].

SGLT2is have shown promise in enhancing plaque stability by targeting these destabilizing factors (Table 4, Figure 3).

Chen et al. [112] investigated the effects of SGLT2is in a diabetic atherosclerotic mouse model specifically designed to mimic plaque instability and rupture. This model was created by inducing tandem stenosis in the right carotid artery, resulting in elevated levels of plaque instability markers, including monocyte chemoattractant protein-1 (MCP-1) and cluster of differentiation 68 (CD68) expression, and an increase in necrotic core size. Mice treated with dapagliflozin exhibited significant plaque stabilization, as indicated by increased collagen content, enhanced fibrosis, improved cap-to-lesion height ratios, and reduced inflammation. Concordantly, Ganbaatar et al. [90] demonstrated that empagliflozin reduced lipid accumulation, macrophage proliferation, and cytokine release, thereby limiting the extent of atherosclerotic lesions in the aortic arch of diabetic apolipoprotein E knockout (ApoE −/−) mice. Additionally, they found that empagliflozin decreased the expression of the macrophage marker CD68, as well as other pro-inflammatory molecules. Moreover, treatment with this agent lowered plasma levels of vasoconstrictive eicosanoids, including prostaglandin E2 and thromboxane B2, leading to vasodilatation and consequently a decrease in shear stress. Furthermore, empagliflozin treatment was associated with an increased ratio of matrix metalloproteinase-2 (MMP-2) to tissue inhibitor of metalloproteinase (TIMP), as well as greater collagen content, both of which contribute to enhanced plaque stability [113]. Lastly, SGLT2 inhibition led to reductions in lipid levels, macrophage accumulation, and neointimal hyperplasia, further supporting the role of SGLT2is in stabilizing atherosclerotic plaques [114,115].

### 4.2. Inflammation Response, Inflammasome, and ROS Production

Inflammation plays a crucial role in the instability of atherosclerotic plaques. Chronic inflammation, driven by immune cells such as macrophages, weakens the fibrous cap of the plaque by releasing enzymes like MMPs. These enzymes degrade the extracellular matrix and collagen fibers within the fibrous cap. As a result, the inflammatory process not only initiates plaque formation but also exacerbates it, driving the plaques toward a vulnerable, unstable state that increases the risk of rupture and serious cardiovascular events [116]. The inflammasome is a critical component of the innate immune system and inflammatory process. It is a multi-protein complex that detects and responds to various physiological or pathological harmful stimuli. The inflammasome is typically activated by pattern recognition receptors (PRRs), which are specialized proteins that recognize specific molecular patterns associated with pathogens or damage-associated molecular patterns (DAMPs) released by stressed or dying cells. The inflammasome can be considered an “immune sensor” that detects danger signals and initiates an inflammatory response to protect the body from infection and injury. However, its dysregulation can lead to excessive inflammation and contribute to disease pathogenesis. Nucleotide-binding domain and leucine-rich repeat protein-3 (NLRP3) is the most well-characterized and extensively studied inflammasome, directly linked to the pathophysiology of several chronic inflammatory diseases, including diabetes and its associated complications [117]. In the context of atherosclerotic plaque formation, significant lipid accumulation triggers the activation of the NLRP3 inflammasome, thereby amplifying the inflammatory response through the production of interleukin-1β (IL-1β) and interleukin-18 (IL-18), which are the primary products of the inflammasome. This heightened inflammation contributes to the progression of plaque instability, ultimately culminating in plaque necrosis [118,119].

Reactive oxygen species (ROS) are highly reactive molecules that are produced by normal cellular metabolism, particularly within mitochondria. They can also be generated in response to various stimuli, such as inflammation, oxidative stress, and endothelial dysfunction. While ROS have some physiological roles, their accumulation and prolonged presence can contribute to the pathogenesis of atherosclerosis and the destabilization of plaques. Hence, high levels of ROS can damage the structural components of plaques, further compromising the stability of their caps.

Many studies have demonstrated that SGLT2is can modulate the inflammation process and NLRP3 inflammasome activation, and consequently ROS production, influencing both the development and stability of atherosclerotic plaques. The anti-inflammatory action of SGLT2 inhibitors has been confirmed by several studies on mice, demonstrating that these drugs can reduce the expression of several molecules involved in inflammation, including MCP-1, VCAM-1, ICAM-1, IL-1β, interleukin-6 (IL-6), and TNF-α [113,114].

In this regard, dapagliflozin demonstrated anti-atherosclerotic properties in a rabbit model by regulating the inflammatory response. It reduced the expression of TNF-α, IL-1β, and IL-6, while also promoting macrophage polarization toward the anti-inflammatory M2 phenotype, regardless of the presence of diabetes [120]. This effect was also confirmed by Xu et al. [121], who observed that SGLT2is may prevent the differentiation of macrophages into the pro-inflammatory M1 phenotype, which is involved in the proliferation and infiltration of plaque-resident macrophages, and may stimulate the polarization of macrophages toward the M2 phenotype. Hodrea et al. [122] provided evidence that dapagliflozin significantly mitigates cardiac inflammation associated with diabetes mellitus. Their findings suggested that dapagliflozin may attenuate inflammatory processes within cardiac tissue under diabetic conditions, highlighting its potential therapeutic role in managing diabetes-related cardiac inflammation beyond its glucose-lowering effects. Moreover, dapagliflozin was shown to reduce inflammasome activation rates, exerting anti-inflammatory and anti-fibrotic effects independently of its glucose-lowering action in type 2 diabetic mice [123,124]. Sukhanov et al. [125] proved that human aortic smooth muscle cells (SMCs) express SGLT2 mRNA. They also found that SGLT2i treatment inhibits NLRP3 expression and reduces oxidative stress, along with the migration and proliferation of SMCs under normal glucose conditions. Specifically, empagliflozin was demonstrated to prevent ROS production stimulated by the proinflammatory cytokine IL-17 and to attenuate inflammatory signaling via NLRP3/caspase-1-dependent mitogenic and migratory proinflammatory cytokines IL-1β and IL-18 in SMCs [125]. Another potential beneficial mechanism of SGLT2is on the inflammasome is related to their ability to lower blood glucose levels. SGLT2is may inhibit the NF-kB signaling pathway through ameliorated glycemic control, which is involved in NLRP3 inflammasome stimulation [126].

Notably, SGLT2is may attenuate the inflammasome by increasing b-hydroxybutyric acid (b-OHB), an endogenous NLRP3 inflammasome inhibitor. This occurs due to their capacity to implement lipolysis and induce ketogenesis [127]. In their study, Birnbaum et al. [128] demonstrated that dapagliflozin effectively reduces NLRP3 inflammasome activation and decreases IL-1β levels, resulting in notable anti-inflammatory and anti-fibrotic effects. Furthermore, Chen et al. [112] demonstrated that dapagliflozin upregulates nicotinamide adenine dinucleotide phosphate oxidase 4 (NADPH oxidase 4), an enzyme involved in ROS production. They also observed that empagliflozin and canagliflozin activate 5′ adenosine monophosphate-activated protein kinase (AMPK), thereby enhancing autophagy in cardiac tissues. Lastly, macrophage autophagy is essential for reducing necrosis in advanced atherosclerotic plaques [129]. This process may be an additional mechanism by which SGLT2is provide protective effects against plaque progression and instability. In conclusion, the activation of the NLRP3 inflammasome significantly contributes to the progression of atherosclerosis by intensifying inflammatory processes within vascular tissues and promoting ROS production, consequently contributing to plaque instability and overall disease advancement. SGLT2is demonstrate the potential to interfere with these pathways, effectively inhibiting NLRP3 inflammasome activity, inflammatory activation, and ROS production through multiple mechanisms. These effects may reduce pro-inflammatory cytokine release and mitigate fibrotic changes within atherosclerotic plaques, thereby stabilizing plaque structure and slowing disease progression (Table 5).

## 5. Myocardial Ischemia and SGLT2 Inhibitors

Following the rupture of an atherosclerotic plaque, multiple pro-thrombotic components are released into the bloodstream, triggering platelet activation, which subsequently promotes thrombus formation and may lead to the occlusion of the affected coronary artery, resulting in myocardial ischemia. Under these conditions, the presence of anemia can further exacerbate the ischemic process by potentiating myocardial oxygen deficit and increasing the cardiac workload, thereby amplifying the severity of ischemic injury. Myocardial ischemia leads to a cascade of pathological events that impair cardiac function. The disruption of normal metabolic pathways under ischemic conditions, including impaired glucose oxidation and enhanced lactate production, plays a critical role in the myocardial energy crisis [130]. Moreover, ischemia severely affects ion homeostasis by disrupting the balance of key electrolytes, such as sodium, potassium, and calcium, which contributes to cell damage [131]. Recent evidence suggests that SGLT2is may offer therapeutic benefits in mitigating the effects of ischemia by modulating these interconnected pathways. These drugs showed benefits in reducing platelet activation, raising hemoglobin levels, improving metabolic efficiency, and maintaining ion balance, making them a potential adjunctive treatment in managing myocardial ischemia (Table 6).

### 5.1. Platelet Activation

Platelets play a critical role in thrombosis, with elevated levels of hyperactive platelets and increased integrin surface expression being associated with a heightened prothrombotic state, particularly in diabetic patients [149]. In this context, SGLT2is have been proposed to exert both direct and indirect antithrombotic effects [149]. Specifically, the administration of empagliflozin has been linked to an improvement in endothelial function, reduction in inflammation, and decreased platelet hyperactivity and aggregation in diabetic fatty rat models [133]. This antithrombotic effect is also reflected in the prevention of upregulation of mRNA expression for pro-thrombotic markers, such as P-selectin and endothelial nitric oxide synthase (eNOS) [133]. Hence, Spigoni et al. [132] observed that SGLT2is, specifically empagliflozin and dapagliflozin, significantly reduced the expression of ADP-induced platelet activation markers, such as CD62P (platelet surface P-selectin) and PAC1 (activated GP IIb/IIIa), when platelets from healthy individuals were exposed to these drugs. These findings suggest a mechanism by which SGLT2is may directly mitigate platelet hyperactivity by modulating platelet activation pathways. Additionally, SGLT2is appear to indirectly inhibit platelet function through multiple pathways, such as reducing ROS generation, restoring NO bioavailability, and suppressing the formation of advanced glycation end-products (AGEs), all of which contribute to a less pro-thrombotic environment [134,135]. These multifaceted effects in inhibiting platelet activation and aggregation highlight the therapeutic potential of SGLT2 inhibition in managing thrombotic risk, particularly in patients with diabetes and associated cardiovascular disease.

### 5.2. Anemia

Anemia is a condition characterized by lower-than-normal hemoglobin levels in the blood, which reduces the blood’s ability to carry oxygen to tissues. In response to anemia, the body compensates by increasing cardiac output, heart rate, and contractility, which raises the heart’s oxygen demand. If coronary artery disease is present, this heightened demand may not be met, with a worsening of ischemic conditions. Moreover, low hemoglobin levels can significantly aggravate the clinical course of acute coronary syndrome (ACS) by intensifying myocardial oxygen deprivation and increasing the heart’s workload, consequently exacerbating ischemic injury. For these reasons, addressing anemia in patients with coronary artery disease is essential for improving cardiac oxygen supply and preventing further ischemic damage.

Several studies have demonstrated that SGLT2is can elevate hematocrit levels in both individuals with and without diabetes mellitus [150,151]. While this effect could be attributed to the diuretic action of these agents, emerging evidence suggests that SGLT2is may directly stimulate erythropoiesis by upregulating renal erythropoietin (EPO) production [136]. In diabetic patients, the introduction of dapagliflozin therapy resulted in an elevation of erythropoietin concentrations, reaching a maximum after 2–4 weeks of treatment [137]. Concurrently, a transient increase in reticulocyte count was observed, followed by a sustained rise in hemoglobin and hematocrit levels [137]. Similarly, Mazer et al. [136] demonstrated that EPO levels significantly increased after one month of empagliflozin treatment in patients with type 2 diabetes mellitus and coronary artery disease. These favorable changes could be due to the SGLT2i-induced effects on fibroblasts in the interstitial tubules, which are responsible for the synthesis of EPO [138]. In patients with diabetes, the proximal tubules become overburdened due to excessive glucose reabsorption, leading to dysfunction in adjacent fibroblasts and a subsequent reduction in erythropoietin production [137,139]. SGLT2is reduce the workload of the proximal tubules and improve tubulointerstitial hypoxia, allowing fibroblasts to resume normal erythropoietin production [140]. However, the mechanism underlying the observed increase in EPO levels in non-diabetic individuals treated with SGLT2is remains to be elucidated [42]. In conclusion, the observed benefits of SGLT2is in ameliorating anemia suggest that these agents may play a significant role in the management of myocardial ischemia, especially in patients with diabetes mellitus and chronic kidney disease. The ability of SGLT2is to address anemia, in addition to their other metabolic effects, may contribute to improved myocardial oxygenation and overall cardiac function.

### 5.3. Modulation of Metabolic Pathways Under Ischemic Conditions

From a biochemical point of view, heart contraction requires the inflow of calcium ions (Ca^2+^) into the cardiomyocyte cytoplasm, where they bind to troponin C myofilaments. This binding triggers the sliding of thin and thick filaments with subsequent activation of a cross-bridge and subsequent cardiac force development. Restarting the contraction cycle requires restoring the ion gradient through active channel pumps such as sodium/calcium exchanger (NCX) or sarcoplasmic reticulum calcium ATPase (SERCA2a) [152]. This process requires high energy consumption, guaranteed by constant adenosine triphosphate (ATP) synthesis. ATP production is provided in mitochondria by oxidative phosphorylation. This biochemical process requires oxygen and reduced forms of coenzymes nicotinamide adenine dinucleotide (NADH) and flavin adenine dinucleotide (FADH2) to generate the proton gradient and, finally, ATP production by ATP synthetase. These reduced coenzymes are produced during the tricarboxylic acid cycle (Krebs cycle), mainly from fatty acyl-coenzyme A (CoA) (50–70%) and pyruvate (20–30%), which are the primary metabolites of fatty acids and carbohydrates, respectively [142]. Free fatty acids (FFAs) can produce more ATP than glucose molecules, although this process requires more oxygen. Interestingly, ketones are even more efficient, producing more energy with less oxygen compared to both glucose and FFAs [153]. To maintain consistent ATP production, cardiomyocytes can use different energy sources depending on substrate availability, workload, and perfusion conditions. During ischemia, whether in chronic or acute situations, limited oxygen availability induces mitochondria to a metabolic switch using fuel that requires less oxygen to produce the same amount of energy. Nevertheless, this adaptative mechanism can lead to cardiomyocyte dysfunction, as happens in heart failure. In this setting, an excessive neuro-hormonal response increases glucose utilization as the primary energy substrate, resulting in intracytoplasmic FFA accumulation, subsequent cardiac steatosis, and lipotoxicity from ROS [154]. SGLT2 receptors are not present in the human heart [155]; nonetheless, SGLT2is show a beneficial effect on cardiomyocytes through various metabolic pathways. Marfella et al. [156] demonstrated that high-glucose conditions in cultured human cardiomyocytes led to increased expression of SGLT2 protein compared to cells exposed to normal glucose levels. Additionally, although the heart does not express SGLT2, it does express SGLT1 protein. SGLT2is are selective for SGLT2 protein, but they exhibit minimal activity against SGLT1 [157]. Under ischemic conditions, there is an upregulation of insulin-independent glucose transporters, specifically GLUT1 and SGLT1, along with the commonly expressed glucose transporters GLUT4 and GLUT8. This upregulation enhances glucose uptake by cardiomyocytes and increases cardioprotection [158]. According to these mechanisms, the observed SGLT2i-induced benefit may be, in part, attributable to the inhibition of SGLT1 receptors [159]. This hypothesis is also supported by a meta-analysis involving a pooled cohort of over 100,000 patients with T2DM treated with SGLT2is, compared to those receiving non-selective SGLT1/2 inhibitors, such as licogliflozin and sotagliflozin. The results indicated that treatment with non-selective inhibitors was associated with a reduced risk of myocardial infarction and stroke, although it also led to an increased incidence of adverse events, including diarrhea and severe hypoglycemia [160]. The inhibition of SGLT1 by SGLT2is in the myocardium could also reduce both sodium and glucose influx while mitigating the hyperglycemia-induced generation of ROS [161]. However, while some SGLT2is can also inhibit SGLT1, the concentrations needed to effectively inhibit SGLT1 are much higher than the plasma levels typically achieved with standard clinical doses of SGLT2is. Despite the above, the primary beneficial effect of SGLT2is on cardiomyocyte metabolism during ischemia is likely mediated through an indirect mechanism. Specifically, SGLT2is induce a “starvation state” that optimizes energy production and reduces oxidative stress [141]. This metabolic shift leads to a re-equilibration of energy substrate utilization by decreasing glucose availability through increased urinary excretion. As a result, lipolysis is promoted and circulating ketone levels rise, independent of diabetic status [162]. Consequently, SGLT2is enhance ATP production during ischemia by increasing the overall availability of energy substrates by approximately 31% [143]. Moreover, they promote more efficient ATP synthesis with reduced oxygen consumption by utilizing ketone bodies—referred to as “super fuel”—instead of glucose or free fatty acids (FFAs) [153].

In support of this, Santos-Gallego et al. [144] conducted a randomized study in which non-diabetic pigs with heart failure induced by myocardial infarction were assigned to either empagliflozin or placebo therapy. Using magnetic resonance imaging and blood sample analysis, they demonstrated that empagliflozin could mitigate adverse cardiac remodeling and heart failure, improve myocardial energetics, and enhance left ventricular systolic function. Moreover, several studies have demonstrated that SGLT2is may increase the activity of sirtuin 1 (SIRT1) and adenosine monophosphate-activated protein kinase (AMPK), while inhibiting anabolic signaling pathways such as the PI3K/Akt/mTOR pathway [163]. These pathways are critical for cellular energy regulation, with AMPK sensing the ratio of ATP to AMP in the cytosol. At the same time, SIRT1 responds to changes in nicotinamide adenine dinucleotide (NAD+) levels, acting as a redox rheostat [145]. By activating SIRT1 and AMPK and suppressing Akt/mTOR signaling, SGLT2is reduce oxidative stress and inflammation and normalize mitochondrial structure and function [83]. Furthermore, SIRT1 activation promotes the stabilization of hypoxia-inducible factor-1α (HIF-1α) under hypoxic conditions, enhancing erythropoietin synthesis and improving oxygen delivery to the myocardium [164]. Moreover, ischemic preconditioning (IPC) upregulates SGLT1 expression through AMPK activity; thus, the inhibition of SGLT1 induced by SGLT2 and SGLT1/2 inhibitors could exacerbate ischemic injury [165]. However, Connelly et al. evaluated the effects of a SGLT1/2 inhibitor compared with a highly selective SGLT2 inhibitor on cardiac function after myocardial infarction [166]. They found that the dual SGLT1/2 inhibitor heightened cardiac dysfunction compared to the selective SGLT2 inhibitor [166]. Thus, SGLT2is could provide a cardioprotective effect in ischemia/reperfusion injury, albeit their effects on SGLT1 inhibition. Similarly, tofogliflozin, a highly sensitive SGLT2i, was demonstrated to not attenuate the cardioprotective effect of IPC in murine models [167].

In conclusion, SGLT2is induce benefits for metabolic pathways by promoting a shift toward efficient energy utilization, enhancing mitochondrial function, reducing oxidative stress, and modulating key regulatory pathways, including SIRT1 and AMPK activation (Table 7). This leads to improved myocardial energetics, reduced cardiac remodeling, and enhanced cardiovascular function, particularly under conditions of ischemia and heart failure.

### 5.4. Modulation of Ion Homeostasis Under Ischemic Conditions

Under ischemic conditions, anaerobic glycolysis becomes the main pathway for ATP synthesis in myocardial cells. This shift leads to the accumulation of lactates and an increase in intracellular H⁺ concentration, leading to cytoplasmic acidosis. Such changes impair the function of ATP-dependent transporters, further reducing ATP production compared to fatty acid β-oxidation [169].

Moreover, the activities of Na⁺/K⁺-ATPase and Ca^2^⁺-ATPase, which normally maintain low cytoplasmic sodium and calcium levels, are disrupted under ischemic conditions, resulting in increased intracellular sodium and calcium concentrations. The elevated intracellular H⁺ concentration due to anaerobic glycolysis further increases the activity of sodium/hydrogen exchanger 1 (NHE1), contributing to the rise in intracellular sodium [170]. As a result of the increased intracellular sodium concentration, the bidirectional sodium/calcium exchanger (NCX) is activated in reverse mode, enhancing calcium overload within the cytoplasm [171]. This inward calcium flux subsequently triggers additional calcium release from the sarcoplasmic reticulum via activation of ryanodine receptor 2 (RyR2), a mechanism known as calcium-induced calcium release [172]. These perturbations in ion homeostasis initiate harmful signaling pathways that promote myocardial injury, hypertrophy, and ultimately lead to cardiac dysfunction [173]. This theory was also confirmed by Wang et al. [174], who demonstrated that NHE1 knockout mice exhibit myocardial ischemia–reperfusion injury tolerance.

Several studies demonstrated the capacity of SGLT2is to modulate these mechanisms, reducing the intracellular concentrations of sodium and calcium. For instance, Ye et al. [146] demonstrated that, in mouse cardiac fibroblasts, dapagliflozin attenuated NHE-1 mRNA elevation by increasing AMP kinase concentrations. This process resulted in a lower myocardial NHE flux, which in turn decreased the intracellular levels of sodium and calcium in the myocardium. These findings were corroborated by Kai Jiang et al. [145] in their study on type 2 diabetic mice following myocardial infarction, demonstrating that empagliflozin significantly reduced infarct size and myocardial fibrosis, with a significant improvement in cardiac function and survival. This effect was mainly attributed to direct inhibition of the activity of the Na^+^/H^+^ exchanger 1 (NHE1) in cardiomyocytes by this agent. Furthermore, Arow et al. [168] found out that the expression of NCX and NHE membrane transporters in rat cardiomyocytes was upregulated following infusion with glucose and/or angiotensin II. By contrast, pre-treatment with dapagliflozin reduced their expression, regardless of glycemic control.

In conclusion, these findings highlight the potential of SGLT2is to modulate key ion transport mechanisms, reducing intracellular sodium and calcium levels, and thereby optimizing myocardial contraction and mitigating pathological myocardial stress and dysfunction under ischemic conditions.

### 5.5. Ischemia/Reperfusion Injury and Oxidative Stress

The role of oxidative stress in atherogenesis and plaque progression/instability has been already discussed above. However, it is important to highlight that ROS production is also a key factor in the cell damage induced by ischemia and eventual blood flow restoration. Under normal conditions, oxygen molecules are utilized by mitochondria in their respiratory chain to produce ATP, and most of this oxygen is ultimately reduced to water. During ischemia, when oxygen availability is limited, mitochondria lack the substrate needed for respiratory chain metabolism, translating in a shift to anaerobic metabolism and a consequent depletion of ATP production. Additionally, electrons from the electron transport chain, which is part of the respiratory chain, accumulate in the mitochondria. To mitigate the accumulation of these electrons, there is a hyperactivation of nicotinamide adenine dinucleotide phosphate oxidase, NADPH oxidase (Nox), which transfers electrons from NADPH to molecular oxygen, leading to ROS production [175]. There are five isoforms of NADPH oxidase, with Nox4 being the most widely expressed isoform in endothelial cells, cardiac myocytes, and fibroblasts [176].

When blood flow is restored and reperfusion occurs, a large amount of oxygen molecules becomes available. However, the cells are in an anabolic metabolism state, and this rapid influx of oxygen can lead to an exacerbation of ROS production. This occurs due to the hyperactivation of NADPH oxidase and the reaction between oxygen molecules and the existing ROS [177]. Moreover, elevated ROS levels can induce the opening of mitochondrial permeability transition pores (MPTPs), leading to the release of cytochrome C and other mitochondrial components into the cytosol, thereby triggering hyper-contracture and promoting apoptotic cell death [178]. Increased ROS concentrations are also a common cause of mitochondrial DNA (mtDNA) damage, activation of apoptotic pathways, fibroblast proliferation, and subsequent progression of maladaptive myocardial remodeling and heart failure [176]. Acting on some of these mechanisms could be a potential strategy to reduce ROS production in the context of ischemia/reperfusion. This was demonstrated by Matsushima et al. [179] on mice with induced myocardial infarction and type 2 diabetes mellitus. This study indicated potential cardiological benefits from using apocynin, an inhibitor of NADPH oxidase activation, compared to placebo. The mice group treated with apocynin showed a decrease in left ventricular (LV) end-diastolic diameter and LV end-diastolic pressure, along with a reduction in interstitial fibrosis of non-infarcted LV.

The role of SGLT2is in these mechanisms has not been extensively investigated. However, Cheng et al. [147] demonstrated in their study that dapagliflozin treatment reduced the expression of NADPH oxidase isoform 4 (Nox4) and subsequently decreased ROS production in mouse lens epithelial cells.

In a recent in vitro study, Ko et al. [148] examined the effects of dapagliflozin, both alone and in combination with sacubitril/valsartan, on rat heart cell cultures exposed to H₂O₂. They found that dapagliflozin treatment significantly reduced both total intracellular and mitochondrial ROS concentrations, as well as mitochondrial permeability transition pore (MPTP) opening. Additionally, dapagliflozin decreased the expression of NADPH oxidase isoforms 1 (Nox1) and 2 (Nox2), along with oxidized proteins, while increasing the expression of antioxidant markers such as SIRT1. This effect was similar in cells treated with sacubitril/valsartan, and it was further amplified when both drugs were combined. Moreover, dapagliflozin treatment significantly reduced the fibrotic and infarcted areas, with even greater reductions when sacubitril/valsartan was added.

In conclusion, these studies collectively support the therapeutic potential of SGLT2is in mitigating oxidative stress during ischemia/reperfusion, thereby improving myocardial function and reducing pathological remodeling in ischemic heart conditions.

Building upon the previously discussed mechanisms, SGLT2 inhibitors reduce ischemic injury through a synergistic mechanism that integrates endothelial function improvement, inflammation suppression, and metabolic adaptation. By enhancing nitric oxide (NO) bioavailability and reducing vascular stiffness, SGLT2is improve coronary perfusion, mitigating oxygen supply–demand mismatch and limiting ischemic burden. Their anti-inflammatory and antithrombotic effects stabilize atherosclerotic plaques, decreasing the likelihood of plaque rupture and thrombosis, which are key triggers of acute ischemic events. Additionally, SGLT2is shift myocardial metabolism toward ketone body utilization, an oxygen-efficient energy source that optimizes ATP production under ischemic conditions. This metabolic adaptation, combined with reduced sodium and calcium overload, protects cardiomyocytes from ischemia–reperfusion injury and myocardial stunning. Together, these effects contribute to improved ischemic tolerance and reduced myocardial injury, supporting the potential role of SGLT2is as adjunctive therapies in ischemic heart disease.

## 6. Conventional and Unconventional Therapeutic Options for Myocardial Ischemia

The management of myocardial ischemia has traditionally relied on pharmacological strategies aimed at mitigating myocardial oxygen demand, enhancing coronary perfusion, and preventing thrombotic complications [180]. Among the cornerstone therapies, beta-blockers play a pivotal role in attenuating heart rate and myocardial contractility, thereby reducing oxygen consumption and myocardial workload [181]. Similarly, lipid-lowering agents, particularly statins, contribute significantly to atherosclerotic plaque stabilization and cardiovascular risk reduction [182]. Angiotensin-converting enzyme (ACE) inhibitors and angiotensin receptor blockers (ARBs) exert protective cardiovascular effects by improving endothelial function, reducing arterial stiffness, and preventing pathological ventricular remodeling [183,184]. Furthermore, antiplatelet agents, including aspirin and P2Y12 inhibitors, are integral to thrombotic risk reduction, playing a crucial role in the secondary prevention of ischemic events [185]. While these pharmacotherapies have demonstrated substantial efficacy in reducing cardiovascular morbidity and mortality, they primarily address symptomatic relief and secondary prevention, often without directly influencing the underlying metabolic and inflammatory pathways that contribute to the progression of ischemic heart disease [180].

In recent years, sodium-glucose cotransporter-2 (SGLT2) inhibitors and glucagon-like peptide-1 receptor agonists (GLP-1 RAs) have emerged as novel therapeutic strategies that offer pleiotropic cardiovascular benefits beyond glucose regulation. Similarly, GLP-1 RAs, such as liraglutide and semaglutide, confer cardioprotective effects through mechanisms that are complementary yet distinct from those of SGLT2 inhibitors [14]. These agents enhance myocardial glucose uptake, reduce oxidative stress, and inhibit cardiomyocyte apoptosis, thereby preserving cardiac function and preventing maladaptive ventricular remodeling [14]. Additionally, GLP-1 RAs have been associated with reductions in systolic and diastolic blood pressure, thereby lowering the incidence of myocardial infarction. Their anti-inflammatory properties further contribute to vascular protection, with evidence suggesting a 16% reduction in nonfatal and total stroke incidence in primary prevention cohorts [186].

While beta-blockers, statins, ACE inhibitors, and antiplatelets remain fundamental in the pharmacological management of myocardial ischemia, their effects are largely centered on hemodynamic modulation, lipid control, and thrombosis prevention, without directly addressing metabolic dysfunction, vascular inflammation, or myocardial bioenergetics. By contrast, SGLT2 inhibitors and GLP-1 RAs exert multifaceted protective effects, improving not only vascular health and atherosclerotic stability but also metabolic efficiency and myocardial resilience to ischemic stress [186]. This broader mechanistic scope underscores their potential to redefine the therapeutic paradigm for ischemic heart disease, extending their benefits beyond diabetic populations. Their integration into cardiovascular treatment guidelines represents a shift toward a more comprehensive and mechanism-driven approach to the prevention and management of myocardial ischemia.

## 7. Future Directions

While SGLT2is demonstrate significant cardiovascular benefits beyond glucose control, several knowledge gaps remain regarding their role in myocardial ischemia. Although evidence supports their ability to reduce atherosclerosis progression, improve endothelial function, and enhance myocardial energy metabolism, further research is needed to establish their direct impact on ischemic heart disease, particularly in non-diabetic populations.

One important step forward would be long-term cohort studies comparing cardiovascular outcomes in patients with coronary artery disease (CAD) treated with SGLT2is versus those receiving conventional therapy alone. Observational studies following patients with stable ischemic heart disease (IHD) or a history of myocardial infarction (MI) could provide valuable insights into whether SGLT2is offer additional protection against recurrent ischemic events. Retrospective analyses using large cardiovascular registries could help to establish whether patients on SGLT2is experience fewer major adverse cardiovascular events (MACEs) compared to those treated with traditional anti-ischemic agents like beta-blockers and ACE inhibitors.

However, to draw definitive conclusions, randomized controlled trials (RCTs) will be necessary. A particularly compelling study would be a head-to-head comparison of SGLT2is versus beta-blockers or ACE inhibitors in patients with stable ischemic heart disease, assessing whether SGLT2is could reduce angina burden, improve exercise tolerance, or lower the need for revascularization procedures. Another avenue of research could explore the effects of early SGLT2i initiation in acute coronary syndrome (ACS) patients, evaluating whether these agents limit infarct size, preserve left ventricular function, or reduce the risk of heart failure post-MI. Additionally, their role in coronary microvascular dysfunction—a growing area of interest, especially in patients with ischemia and no obstructive coronary disease (INOCA)—remains largely unexplored. A clinical trial assessing SGLT2is in patients with microvascular angina could help to determine whether their vasoprotective effects extend to small vessel disease, which currently lacks effective targeted therapies.

Beyond clinical outcomes, unresolved mechanistic questions still need to be addressed. The mechanisms through which SGLT2is influence plaque composition and stability, as well as their role in modulating myocardial ischemia–reperfusion injury, remain unclear. Moreover, the impact of SGLT2is on coronary microvascular dysfunction, a key contributor to ischemia in patients without significant epicardial coronary obstruction, remains largely unexplored.

Further mechanistic studies should aim to clarify how SGLT2is interact with inflammatory, oxidative stress, and metabolic pathways to confer cardioprotection. Additionally, real-world data and long-term follow-up studies are essential to assess their safety, cost effectiveness, and clinical applicability across diverse cardiovascular populations.

Addressing these gaps through randomized controlled trials and translational research will be crucial for defining the optimal role of SGLT2is in ischemic heart disease, potentially expanding their use beyond current guidelines and shaping a new, mechanism-driven approach to cardiovascular therapy.

## 8. Conclusions

Sodium-glucose cotransporter 2 inhibitors (SGLT2is) demonstrate significant potential in ameliorating myocardial ischemia through a multifaceted approach. These agents act at multiple stages of ischemic pathophysiology, beginning with atherogenesis by controlling key risk factors such as hypertension, hyperuricemia, dyslipidemia, diabetes, and homocysteinemia, which collectively contribute to plaque formation. Additionally, SGLT2is improve vascular function by mitigating endothelial dysfunction and reducing arterial stiffness, thus enhancing overall coronary flow. Furthermore, SGLT2is may reduce plaque vulnerability and rupture by modulating plaque composition and inflammation, thereby preventing thrombotic events. In the context of myocardial ischemia, SGLT2is attenuate platelet activation, alleviate anemia-induced oxygen deprivation, and regulate critical metabolic pathways and ion homeostasis, all of which are disrupted under ischemic conditions. In conclusion, SGLT2is could attenuate atherogenesis and thrombotic events, while improving vascular and myocardial health, thus representing a promising therapeutic option for the treatment of myocardial ischemia.

However, further clinical studies are essential to confirm their efficacy in these broader indications, assess their long-term benefits, and better understand the full extent of their clinical applicability in diverse patient populations.

## Figures and Tables

**Figure 1 ijms-26-02103-f001:**
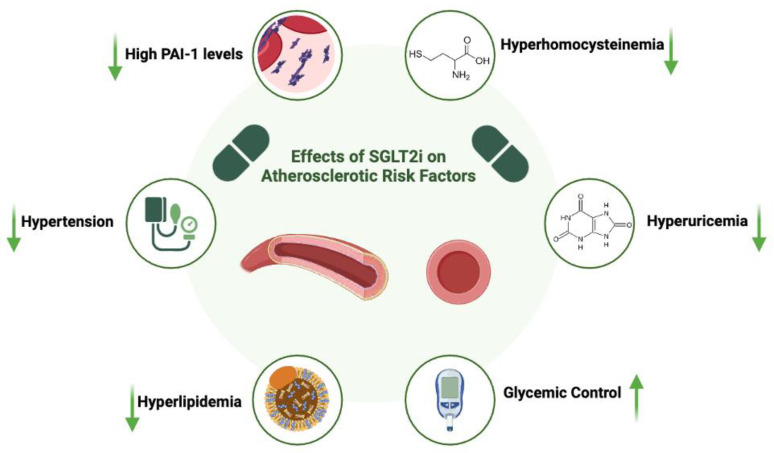
The pleiotropic effects of SGLT2 inhibitors on atherosclerosis pathogenesis. Created in BioRender. Lanciotti, M. (2025) https://BioRender.com/y25t695 (accessed on 20 February 2025). SGLT2i = Sodium-Glucose Cotransporter-2 Inhibitor; PAI-1 = Plasminogen Activator Inhibitor-1; ↑ = raised; ↓ = reduced.

**Figure 2 ijms-26-02103-f002:**
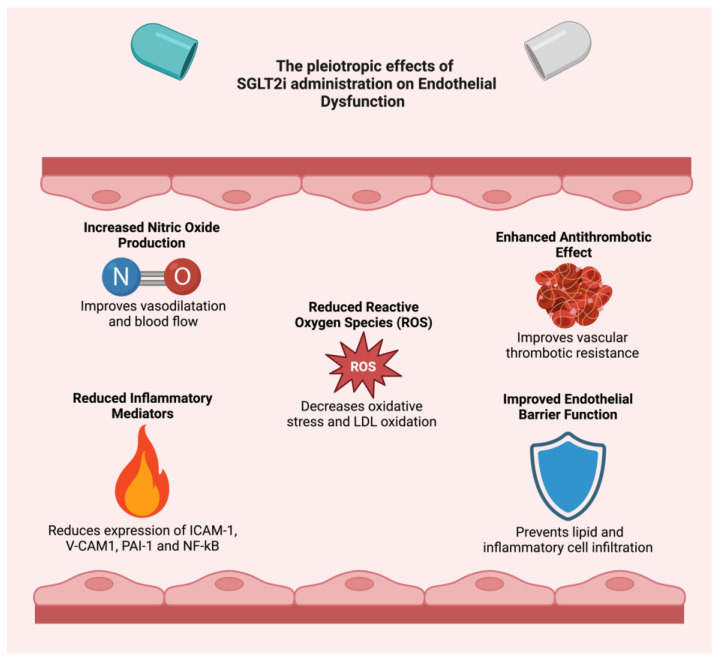
Effects of SGLT2 inhibitors on endothelial dysfunction. Created in BioRender. Lanciotti, M. (2025) https://BioRender.com/z48f832 (accessed on 20 February 2025).

**Figure 3 ijms-26-02103-f003:**
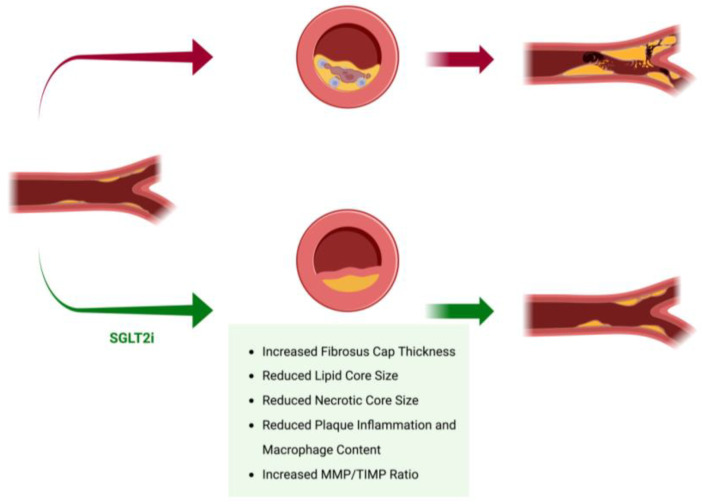
Modulatory effects of SGLT2 inhibitors on atherosclerotic plaque stability and composition. Created in BioRender. Lanciotti, M. (2025) https://BioRender.com/l85b701 (accessed on 20 February 2025). Legend. MMP = Matrix Metalloproteinase; TIMP = Tissue Inhibitor of Metalloproteinase.

**Table 1 ijms-26-02103-t001:** Effects of SGLT2is on key cardiovascular risk factors associated with atherosclerosis.

Risk Factor	Effects of SGLT2 Inhibitors
Hypertension	↓ systolic and diastolic BP [18,19,20] Sustain BP reduction long-term [20] Maintain circadian BP rhythm, reducing nocturnal BP [22,23,24] ↓ BP without increasing heart rate (through lowering SNS activity) [27,28,29,30]
Diabetes	↓ blood glucose through glucosuria Improve insulin sensitivity
Hyperlipidemia	Modulate lipid metabolism enhancing fat utilization [4,45,46] ↓ hepatic lipogenesis, mitigate hepatic steatosis, and stimulate lipolysis [4,45,47]
Hyperuricemia	↓ serum uric acid levels by increasing renal excretion through osmotic diuresis and competitive inhibition of renal urate transporters [53,54,55,56,57]
Hyper-homocysteinemia	↓ homocysteine levels likely through improved insulin sensitivity, decreased inflammation, and enhanced renal function [59]
High PAI-1 Levels	↓ PAI-1 levels by improving insulin sensitivity, reducing inflammation, improving renal function, and promoting weight loss [61,62]

Abbreviations: SGLT2i = Sodium-Glucose Cotransporter-2 Inhibitor; BP = Blood Pressure; SNS = Sympathetic Nervous System; PAI-1 = Plasminogen Activator Inhibitor-1; ↓ = reduce.

**Table 2 ijms-26-02103-t002:** Effects of SGLT2is on endothelial dysfunction and vascular health.

Endothelial Dysfunction Mechanism	Effects of SGLT2 Inhibitors
Nitric Oxide (NO) Production	↑ NO bioavailability, improving vasodilation and reducing blood pressure [82] Activate the SIRT1/eNOS signaling pathway to enhance endothelial function [82]
Inflammatory Regulation	↓ expression of inflammatory and adhesion molecules (ICAM-1, VCAM-1, PAI-1, NF-κB), decreasing macrophage infiltration and slowing atherosclerosis progression [71]
Reactive Oxygen Species (ROS) Production	↓ ROS production, mitigating oxidative stress and endothelial damage, thus preventing LDL oxidation and atherosclerotic plaque progression [82,83]
Barrier Function	Preserve endothelial barrier integrity, limiting lipid and inflammatory cell infiltration into the vessel walls [84,85]
Antithrombotic Effect	Restore the antithrombotic properties of the endothelium [71]
Vasorelaxation	↑ endothelium-independent vasorelaxation, enhancing blood flow and reducing vascular resistance [71] ↑ endothelium-dependent vasorelaxation, enhancing micro-vessel density and tissue perfusion [84,85]
Vascular Smooth Muscle Cell (VSMC) Proliferation	↓ proliferation and migration of VSMCs, preventing the thickening of the vessel walls [88,89]
Microvascular Health	↑ microvascular health by promoting eNOS phosphorylation, improving micro-vessel density, and enhancing tissue perfusion [82,83,84,85]
Cellular Aging	↓ cellular aging and improve endothelial function by modulating the SIRT1/AMPK pathway, lowering oxidative stress and inflammatory responses [83]

Abbreviations: SGLT2i = Sodium-Glucose Cotransporter-2 Inhibitor; NO = Nitric Oxide; eNOS = Endothelial Nitric Oxide Synthase; SIRT1 = Sirtuin 1; ICAM-1 = Intercellular Adhesion Molecule 1; VCAM-1 = Vascular Cell Adhesion Molecule 1; PAI-1 = Plasminogen Activator Inhibitor-1; NF-κB = Nuclear Factor Kappa-light-chain-enhancer of Activated B cells; ROS = Reactive Oxygen Species; LDL = Low-Density Lipoprotein; VSMCs = Vascular Smooth Muscle Cells; AMPK = AMP-Activated Protein Kinase; ↓ = reduce; ↑ = raise.

**Table 3 ijms-26-02103-t003:** Impact of SGLT2is on arterial stiffening and vascular health.

Mechanisms of Arterial Stiffening	Effects of SGLT2 Inhibitors
Chronic Inflammation and Vascular Calcification	↓ inflammatory markers and oxidative stress, slowing vascular calcification and arterial stiffening [103]
Medial Degeneration	↓ medial degeneration, improving vascular elasticity [103]
Endothelial Dysfunction	↑ endothelial function, leading to better vascular relaxation and reduced stiffness
Vascular Smooth Muscle Relaxation	↑ potassium channels and protein kinase G (PKG) signaling pathways, promoting vasodilation and reducing arterial stiffness [104]
Impact on Comorbid Conditions	↓ vascular aging and arterial stiffness by addressing multiple cardiovascular risk factors (blood pressure, glycemic control, weight loss)
Pulse Pressure	↓ central systolic blood pressure and pulse pressure, improving arterial stiffness [105,106,107,108,109]

Abbreviations: SGLT2i = Sodium-Glucose Cotransporter-2 Inhibitor; PKG = protein kinase G; ↓ = reduce; ↑ = raise.

**Table 4 ijms-26-02103-t004:** Effects of SGLT2 inhibitors on atherosclerotic plaque composition.

Plaque Feature	Effects of SGLT2 Inhibitors
Fibrous Cap Thickness	↑ collagen content [112] ↑ fibrosis [112]
Lipid Core Size	↓ lipid accumulation [90,113]
Necrotic Core Size	↓ necrotic core size [112]
Plaque Inflammation	↓ inflammation (MCP-1, IL-6, cytokines) [90,113]
Macrophage Content (CD68 Expression)	↓ macrophage proliferation [90,113,114] ↓ pro-inflammatory markers (CD68) [90,113]
Matrix Metalloproteinase (MMP)/TIMP Ratio	↑ MMP-2/TIMP ratio, favoring plaque stabilization [113]

Abbreviations: MCP-1 = Monocyte chemoattractant protein-1; CD68 = Cluster of Differentiation 68 (Macrophage marker); MMP-2 = Matrix Metalloproteinase-2; TIMP = Tissue Inhibitor of Metalloproteinase; IL-6 = Interleukin-6; ↓ = reduce; ↑ = raise.

**Table 5 ijms-26-02103-t005:** Positive effects of SGLT2is on inflammation and ROS production in the context of atherosclerotic plaque stability.

Mechanisms	Effects of SGLT2 Inhibitors	Study Findings
Reduction in Inflammatory Cytokines	↓ pro-inflammatory cytokines	↓ TNF-α, IL-1β, and IL-6 in animal models, key drivers of plaque instability [113,114,120]
Macrophage Polarization	Shift toward anti-inflammatory M2 phenotype	↑ macrophage polarization toward the M2 phenotype, mitigating plaque inflammation and instability [120,121]
NLRP3 Inflammasome Inhibition	↓ NLRP3 inflammasome activation [114,117]	↓ NLRP3 expression in aortic smooth muscle cells (SMCs), reducing inflammatory activation [125]
Oxidative Stress	↓ ROS production	↓ oxidative stress, stabilizing plaque structures [112]
Blood Glucose Control	↑ glycemic control, reducing inflammasome activation	↓ the NF-kB signaling pathway, reducing NLRP3 inflammasome activation linked to high blood glucose [126]
Ketogenesis and Lipolysis	↑ production of β-hydroxybutyric acid (β-OHB)	↑ ketogenesis, which inhibits NLRP3 inflammasome activation via β-OHB, leading to reduced inflammation [127]
Autophagy Enhancement	↑ autophagy, reducing necrosis in plaques	↑ AMPK and enhance macrophage autophagy, reducing plaque necrosis [120,129]

Abbreviations: SGLT2i = Sodium-Glucose Cotransporter-2 Inhibitor; TNF-α = Tumor Necrosis Factor Alpha; IL-1β = Interleukin-1 Beta; IL-6 = Interleukin-6; M2 = Anti-inflammatory Macrophage Phenotype; NLRP3 = NOD-Like Receptor Pyrin Domain Containing 3; ROS = Reactive Oxygen Species; NF-κB = Nuclear Factor Kappa-light-chain-enhancer of Activated B cells; β-OHB = Beta-Hydroxybutyric Acid; AMPK = AMP-Activated Protein Kinase; SMCs = Smooth Muscle Cells; ↓ = reduce; ↑ = raise.

**Table 6 ijms-26-02103-t006:** Effects of SGLT2is in the amelioration of myocardial ischemia.

Mechanism/Pathway	Effects of SGLT2 Inhibitors	Clinical/Experimental Evidence
Platelet Activation and Thrombotic Risk	↓ platelet hyperactivity and aggregation, thus preventing thrombosis	↓ platelet activation markers (CD62P, PAC1) in healthy individuals [132] ↓ platelet aggregation in diabetic rat models [133] ↑ endothelial function and reduce P-selectin, eNOS expression in models of thrombosis [133] ↓ indirectly platelet activation/aggregation by reducing ROS, restoring NO, and suppressing AGEs formation [134,135]
Anemia and Oxygenation	↑ erythropoietin (EPO), raising hematocrit levels and improving oxygen delivery	↑ EPO levels in diabetic and non-diabetic models, improving myocardial oxygen supply [42,136,137,138,139,140]
Metabolic Pathways in Ischemia	↑ metabolic shift toward ketone bodies, reducing oxygen demand.	↑ ketone production and enhanced ATP synthesis with reduced oxygen consumption [141,142,143] ↑ mitochondrial efficiency in ischemic myocardium [144]
Ion Homeostasis in Ischemia	↓ intracellular sodium and calcium, preventing ion overload and cell damage.	↓ sodium and calcium accumulation in cardiac cells in ischemic models [145] ↑ ion homeostasis in ischemic myocardium by inhibiting NHE1 activity [145,146]
Oxidative Stress and Reperfusion Injury	↓ ROS production and mitigation of mitochondrial damage	↓ Nox4 expression and ROS in ischemic cardiac models [147] ↓ MPTP opening and oxidative damage in rat heart cells [148]
Improvement in Myocardial Function	Enhance myocardial energetics, improving cardiac output and function	↑ left ventricular systolic function in ischemic heart failure models [144]
Cardiac Remodeling and Fibrosis	↓ infarct size, fibrosis, and maladaptive remodeling	↓ myocardial infarct and fibrosis in ischemic models [145]

Abbreviations: SGLT2i = Sodium-Glucose Cotransporter-2 Inhibitor; CD62P = Platelet Surface P-Selectin; PAC1 = Activated Glycoprotein IIb/IIIa; eNOS = Endothelial Nitric Oxide Synthase; ROS = Reactive Oxygen Species; NO = Nitric Oxide; AGEs = Advanced Glycation End-products; EPO = Erythropoietin; ATP = Adenosine Triphosphate; ROS = Reactive Oxygen Species; MPTP = Mitochondrial Permeability Transition Pore; Nox = NADPH Oxidase; NHE1 = Sodium/Hydrogen Exchanger 1; Nox4 = NADPH Oxidase 4; ↓ = reduce; ↑ = raise.

**Table 7 ijms-26-02103-t007:** Key molecular targets and pathways modulated by SGLT2is in myocardial ischemia.

Molecular Target/Pathway	Effects of SGLT2 Inhibitors	Clinical/Experimental Evidence
NADPH Oxidase (Nox4)	↓ Nox4 expression	↓ Nox4 levels in animal models, leading to a decrease in ROS production [147]
AMPK and SIRT1 Activation	Activate AMPK and SIRT1	↑ AMPK and SIRT1 activity, leading to reduced ROS and improved mitochondrial efficiency [163]
NHE1 and NCX Inhibition	↓ intracellular sodium and calcium overload, optimizing ion homeostasis	↓ NHE1 mRNA levels in mouse cardiac fibroblasts [146] ↓ NHE1 and NCX expression in rat cardiomyocytes [168] ↓ calcium and sodium influx, improving ischemic conditions [145,146]
Ketone Body Production	↑ ketone body production, reducing oxygen demand and promoting efficient ATP synthesis	↑ a shift to ketone metabolism, enhancing myocardial ATP production during ischemia [141,143,162]
Inflammation and Endothelial Function	↓ inflammation and ↑ endothelial function, supporting cardiac repair	↓ inflammatory markers and ↑ endothelial function in ischemic models [147,148]

Abbreviations: Nox4 = NADPH Oxidase Isoform 4; AMPK = Adenosine Monophosphate-Activated Protein Kinase; SIRT1 = Sirtuin 1; NHE1 = Sodium/Hydrogen Exchanger 1; NCX = Sodium/Calcium Exchanger; EPO = Erythropoietin; ROS = Reactive Oxygen Species; MPTP = Mitochondrial Permeability Transition Pore; GLUT = Glucose Transporter; PI3K/Akt/mTOR = Phosphoinositide 3-Kinase/Protein Kinase B/Mammalian Target of Rapamycin Pathway; HIF-1α = Hypoxia-Inducible Factor 1 Alpha; ↓ = reduce; ↑ = raise.

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
