# Peer review of "Sodium-Glucose Transporter-2 Inhibitors (SGLT2i) and Myocardial Ischemia: Another Compelling Reason to Consider These Agents Regardless of Diabetes"

_ijms, 2025, doi:10.3390/ijms26052103_

Round 1
Reviewer 1 Report
Comments and Suggestions for Authors
The review frequently highlights the SIRT1/eNOS signaling pathways, which is appropriate. However, a more detailed examination of their clinical relevance and a comparison with other pathways, such as AMPK, could enhance the discussion.
The hyperlipidemia section could be expanded to address conflicting evidence surrounding LDL cholesterol levels more thoroughly.
Consider including a brief discussion of other therapeutic options for myocardial ischemia to better position the distinctive advantages of SGLT2 inhibitors. Moreover in order to empower the discussion around the pleiotropic effects of SGTL2i authors should include their antiarrhythmic characteristics (doi: 10.1111/jce.16344.)
A “Future Directions” section would be a valuable addition, summarizing existing knowledge gaps, proposing potential clinical trials, and identifying unresolved research questions.
Several minor grammatical issues, such as incorrect comma placement and omitted articles, were noted throughout the manuscript.
Author Response
The review frequently highlights the SIRT1/eNOS signaling pathways, which is appropriate. However, a more detailed examination of their clinical relevance and a comparison with other pathways, such as AMPK, could enhance the discussion.
We appreciate the feedback from the reviewer, thus we added more information regarding these signaling pathways, as suggested (Page 7, Lines 306-316).
The hyperlipidemia section could be expanded to address conflicting evidence surrounding LDL cholesterol levels more thoroughly.
We thank the reviewer for the comment. We expanded the section as suggested (Page 5, Lines 187-198)
Consider including a brief discussion of other therapeutic options for myocardial ischemia to better position the distinctive advantages of SGLT2 inhibitors.
We thank the reviewer for his/her suggestion. We added a new chapter called “Conventional and unconventional therapeutic options for myocardial ischemia”, discussing about the established therapeutic options for myocardial ischemia and the potential role of SGLT2i (page 21, Lines 881-919)
Moreover in order to empower the discussion around the pleiotropic effects of SGTL2i authors should include their antiarrhythmic characteristics (doi: 10.1111/jce.16344.)
We thank the reviewer for the comment. We briefly discussed beneficial effects on arrhythmias in Introduction section (Page 2, Lines 50-57).
A “Future Directions” section would be a valuable addition, summarizing existing knowledge gaps, proposing potential clinical trials, and identifying unresolved research questions.
We thank the reviewer. As suggested, we added a new section (“7. Future Directions”) that outlines existing knowledge gaps, proposes potential clinical trials to evaluate the role of SGLT2i in myocardial ischemia, and highlights unresolved research questions regarding their impact on plaque stability, ischemia-reperfusion injury, and coronary microvascular dysfunction (Page 22, Lines 921-933; Page 23, Lines 934-962)
Several minor grammatical issues, such as incorrect comma placement and omitted articles, were noted throughout the manuscript.
We thank the reviewer for the comment. We did our best to improve the quality of manuscript.
Reviewer 2 Report
Comments and Suggestions for Authors
This manuscript provides a thorough and well-structured review of the cardiovascular benefits of sodium-glucose transporter-2 inhibitors (SGLT2i) in the context of myocardial ischemia, independent of their glycemic effects.
The manuscript does a great job summarizing the benefits of SGLT2i on endothelial function, inflammation, and myocardial metabolism. However, a clearer mechanistic pathway linking these effects to reduced ischemic injury would strengthen the argument.
While the manuscript focuses on SGLT2 inhibitors, it is important to also briefly mention the potential cardiovascular benefits of GLP-1 receptor agonists (GLP-1RAs), as they have been shown to provide complementary mechanisms in cardiovascular protection. I suggest adding a small paragraph discussing the role of GLP-1RAs in myocardial ischemia, atherosclerosis, and heart failure. A recent review (DOI: 10.2174/0113816128304097240529053538) extensively addresses the dual impact of these agents, and I recommend incorporating this reference into the discussion.
Discussing where SGLT2i stand in current cardiovascular treatment algorithms would provide additional clinical relevance.
The discussion on ischemic preconditioning and metabolic effects of SGLT2i could be expanded with more clinical trial data.
The conclusion should reinforce the clinical applicability of SGLT2i beyond diabetes and heart failure, summarizing key findings concisely.
Ensure that abbreviations are consistently defined when first introduced.
The reference list should be updated with more recent literature.
Author Response
The manuscript does a great job summarizing the benefits of SGLT2i on endothelial function, inflammation, and myocardial metabolism. However, a clearer mechanistic pathway linking these effects to reduced ischemic injury would strengthen the argument.
We thank the reviewer for his/her suggestion. As recommended, we clarified the mechanistic pathway linking SGLT2i to reduced ischemic injury by integrating their effects on endothelial function, inflammation suppression, and metabolic adaptation into a cohesive framework (Page 21, lines 865-878).
While the manuscript focuses on SGLT2 inhibitors, it is important to also briefly mention the potential cardiovascular benefits of GLP-1 receptor agonists (GLP-1RAs), as they have been shown to provide complementary mechanisms in cardiovascular protection. I suggest adding a small paragraph discussing the role of GLP-1RAs in myocardial ischemia, atherosclerosis, and heart failure. A recent review (DOI: 10.2174/0113816128304097240529053538) extensively addresses the dual impact of these agents, and I recommend incorporating this reference into the discussion.
We thank the reviewer for his/her suggestion. Specifically, we did not mention GLP-1RAs as our review is focused on SGLT2i. A clear and extensive evaluation of cardiovascular benefits related to new anti-diabetic drugs (including GLP1-RAs and DPP4i) outreaches the aim of this manuscript. Similarly, a brief and concise paragraph will not result exhaustive, considering the important role of these agents. However, we mentioned GLP1-RAs in the new “Conventional and unconventional therapeutic options for myocardial ischemia” paragraph, adding the suggested reference (page 21, Lines 881-919).
Discussing where SGLT2i stand in current cardiovascular treatment algorithms would provide additional clinical relevance.
We thank the reviewer for the feedback. As recommended, we have expanded our discussion on the position of SGLT2 inhibitors (SGLT2i) within current cardiovascular treatment algorithms, emphasizing their established role in heart failure management, as reflected in European Society of Cardiology (ESC) guidelines. Additionally, we have highlighted the absence of SGLT2i in guideline-directed therapy for coronary artery disease (CAD) and myocardial ischemia (Page 2, Lines 46-49; Page 22, Lines 909-919).
The discussion on ischemic preconditioning and metabolic effects of SGLT2i could be expanded with more clinical trial data.
We thank the reviewer for the comment. However, few data are available regarding the effects of SGLT2i on ischemic preconditioning (IP), mainly related to pre-clinical or murine studies. Thus, we briefly discussed current knowledges on IP and SGLT2i (Page 19, Lines 751-760).
The conclusion should reinforce the clinical applicability of SGLT2i beyond diabetes and heart failure, summarizing key findings concisely.
We thank the reviewer for his/her suggestion. As recommended, we expanded the "Conclusion" section to concisely summarize the key findings of our review, emphasizing the clinical applicability of SGLT2i beyond diabetes and heart failure. We reinforced the potential role of SGLT2i in cardiovascular disease management, particularly in coronary artery disease (CAD) and myocardial ischemia, while also addressing current knowledge gaps and the need for further clinical investigation.
Ensure that abbreviations are consistently defined when first introduced.
We thank the reviewer for his/her suggestion. We defined abbreviations when first mentioned.
The reference list should be updated with more recent literature.
We thank the reviewer for the comment. We added more recent references.
Round 2
Reviewer 1 Report
Comments and Suggestions for Authors
Congratulations to the authors for having significantly improved their manuscript.
Reviewer 2 Report
Comments and Suggestions for Authors
The manuscript has been appropriately revised
Comments on the Quality of English LanguageNo apparent issues